# Insulin as Monotherapy and in Combination with Other Glucose-Lowering Drugs Is Related to Increased Risk of Diagnosis of Pneumonia: A Longitudinal Assessment over Two Years

**DOI:** 10.3390/jpm11100984

**Published:** 2021-09-29

**Authors:** Michael Leutner, Michaela Kaleta, Luise Bellach, Alexander Kautzky, Stefan Thurner, Peter Klimek, Alexandra Kautzky-Willer

**Affiliations:** 1Department of Internal Medicine III, Clinical Division of Endocrinology and Metabolism, Medical University of Vienna, Waehringer Guertel 18-20, A-1090 Vienna, Austria; michael.leutner@meduniwien.ac.at (M.L.); luise.bellach@meduniwien.ac.at (L.B.); 2Section for Science of Complex Systems, CeMSIIS, Medical University of Vienna, Spitalgasse 23, A-1090 Vienna, Austria; kaleta@csh.ac.at (M.K.); stefan.thurner@meduniwien.ac.at (S.T.); peter.klimek@meduniwien.ac.at (P.K.); 3Complexity Science Hub Vienna, Josefstaedter Strasse 39, A-1080 Vienna, Austria; 4Department of Psychiatry and Psychotherapy, Medical University of Vienna, A-1090 Vienna, Austria; alexander.kautzky@meduniwien.ac.at; 5Santa Fe Institute, 1399 Hyde Park Road, Santa Fe, NM 85701, USA; 6Gender Institute, A-3571 Gars am Kamp, Austria

**Keywords:** diabetes mellitus, pneumonia, antidiabetic drugs

## Abstract

Objective: Patients with type 2 diabetes mellitus (T2DM) are at an increased risk of developing infectious diseases such as pneumonia. Hitherto, there has been uncertainty as to whether there is a relationship between different antidiabetic drug combinations and development of pneumonia in this specific cohort. Research Design and Methods: In this longitudinal retrospective study we used multiple logistic regression analysis to assess the odds ratios (ORs) of pneumonia during an observational period of 2 years in 31,397 patients with T2DM under previously prescribed stable antidiabetic drug combinations over a duration of 4 years in comparison to 6568 T2DM patients without drug therapy over 4 years adjusted for age, sex and hospitalization duration. Results: Of the 37,965 patients with T2DM, 3720 patients underwent stable monotherapy treatment with insulin (mean age: 66.57 ± 9.72 years), 2939 individuals (mean age: 70.62 ± 8.95 y) received stable statin and insulin therapy, and 1596 patients were treated with a stable combination therapy of metformin, insulin and statins (mean age: 68.27 ± 8.86 y). In comparison to the control group without antidiabetic drugs (mean age: 72.83 ± 9.96 y), individuals undergoing insulin monotherapy (OR: 2.07, CI: 1.54–2.79, *p* < 0.001); insulin and statin combination therapy (OR: 2.24, CI: 1.68–3.00, *p* < 0.001); metformin, insulin and statin combination therapy (OR: 2.27, CI: 1.55–3.31, *p* < 0.001); statin, insulin and dipeptidyl peptidase-4 inhibitor (DPP-IV inhibitor) combination therapy (OR: 4.31, CI: 1.80–10.33, *p* = 0.001); as well as individuals treated with metformin and sulfonylureas (OR: 1.70, CI: 1.08–2.69, *p* = 0.02) were at increased risk of receiving a diagnosis of pneumonia. Conclusions: Stable monotherapy with insulin, but also in combination with other antidiabetic drugs, is related to an increased risk of being diagnosed with pneumonia during hospital stays in patients with type 2 diabetes mellitus compared to untreated controls.

## 1. Introduction

In the general population, the number of patients with pneumonia is on the rise [1], and healthcare costs are thus increasing dramatically. Diabetes mellitus is a metabolic disease leading to an increased risk of infections due to the interplay between vascular complications and reduced defense capacities of the immune system [2,3]. In general, patients with type 2 diabetes are at an increased risk of several complications [4], including infectious diseases such as pneumonia [5,6]. It is also known that a worse glycemic profile can further aggravate this risk, as well as increase the mortality rate [7,8]. Diabetes may lead to an impairment of immune cells directly [9,10] or indirectly via reduced transport of immune cells to the site of injury or infection due to advanced vascular damage [11]. Serious respiratory infections are especially relevant in diabetic patients, since diabetes mellitus can lead to poorer outcomes in lung diseases via multiple pathophysiological mechanisms such as autonomous neuropathy or phrenic nerve neuropathy [12], as well as impairment of the immune system [3]. Therefore, it is warranted to deepen the understanding of the interactions of antidiabetic drugs and respiratory infections—especially considering that diabetics also have a poor outcome in the context of the SARS-CoV-2 outbreak [13], likely due to more comorbidities [14] and increased susceptibility to infection [3]. With further complicating elements like seasonal flu and pneumonia, it is pivotal to search for readily available and cost-effective treatment options to reduce the risk of pneumonia in diabetics. However, studies investigating the role of different antidiabetic drug combinations in the development of respiratory infections are sparse. For example, statin therapy has been connected to better pneumonia-related outcomes [15]. Hence, the aim of this study is to compare the impact of different stable antidiabetic drug combinations and monotherapies on the outcome of pneumonia in patients with type 2 diabetes mellitus (T2DM) during an observational period of 6 years.

## 2. Results

### 2.1. Baseline Characteristics

Table 1 shows the baseline characteristics of patients in the therapy groups and the control cohort. We observe that the therapy groups receiving insulin monotherapy, insulin therapy combined with statins, biguanides combined with insulin and statins or therapy with DPP-IV inhibitors combined with insulin and statins show higher relative numbers of patients with the outcome of pneumonia during the observational period when compared to other antidiabetic drug therapies. Although we could not find a higher age in these high-risk groups, the patients in these groups had an increased risk of being diagnosed with several comorbidities (see Appendix A).

To test for the robustness of the regression results, we performed two additional analyses. First, we implemented an identical analysis without the inclusion of statin medication (only antidiabetic ATC A10 codes). Second, we specifically adjusted for comorbidities as binary covariates in the regression model. There was no qualitative change in the observed effects.

### 2.2. Relationship between Different Types of Glucose Lowering Drugs and Pneumonia

As presented in Figure 1, insulin therapy was in general related to an increased risk of pneumonia in individuals with type 2 diabetes mellitus. In detail, patients receiving insulin monotherapy (OR: 2.07, CI: 1.54–2.79, *p* < 0.001), insulin + statins (OR:2.24, CI: 1.68–3.00, *p* < 0.001), insulin + DPP-IV inhibitors + statins (OR: 4.31, CI: 1.80–10.33, *p* = 0.001), insulin + metformin + statins (OR:2.27, CI: 1.55–3.31, *p* < 0.001) or a combination therapy of sulfonylureas + metformin (OR:1.70, CI: 1.08–2.69, *p* = 0.02) were at increased risk of being diagnosed with pneumonia during the observational period. Interestingly, metformin, either as monotherapy or in combination with other glucose lowering drugs, was not related to a reduced risk of pneumonia in the present study.

### 2.3. Sex-Specific Analysis

In a sex-specific analysis, insulin monotherapy was related to an increased pneumonia risk in both males (2.02, CI: 1.35–3.03, *p* < 0.001) and females (OR: 2.17, CI: 1.40–3.37, *p* < 0.001). As shown in Table 2 and Table 3, the combination of insulin and statins and the combination of insulin, metformin and statins were also related to increased risk of pneumonia in both sexes. Interestingly, the combination therapy of sulfonylureas and metformin showed sex-specific results only in females and was related to an increased OR for pneumonia (OR: 1.91, CI: 1.00–3.64, *p* = 0.048).

## 3. Discussion

In view of the increased risk of infections in patients with type 2 diabetes mellitus, it is of the utmost importance to investigate the risk of pneumonia under different antidiabetic drug combinations. In this nationwide population-based study including 37,965 patients with type 2 diabetes mellitus, we could show that over a study period of 6 years, insulin therapy was related to an increased risk of pneumonia in patients with type 2 diabetes mellitus. Specifically, patients treated with insulin monotherapy, but also in combination with statins, metformin or DPP-IV inhibitors, had an increased risk of diagnosis of pneumonia when compared to individuals without antidiabetic drug therapy.

Due to their impaired immune system [3,9], type 2 diabetic patients are at increased risk of contracting pneumonia [16]. However, the role of antidiabetic drugs in the risk of developing pneumonia in type 2 diabetic patients has not been clarified at all so far. Recently, aggravation of systemic inflammation and increased mortality rate and organ injuries in patients infected with SARS-CoV2 also receiving insulin treatment for type 2 diabetes mellitus has been reported in a study by Yu et al. [17]. The increased risk associated with insulin therapy has also been demonstrated in a study by Cariou et al., who showed that in patients with type 2 diabetes who were hospitalized with a SARS-CoV-2 infection, insulin therapy was related to an increased mortality rate [18]. Similar results have also been found in an analysis of the entire Scottish population, which found that patients with SARS-CoV-2 undergoing treatment with insulin or sulfonylureas had the greatest risk of an unfavorable outcome, including a higher risk of being hospitalized in an intensive-care unit [19]. In a nationwide observational cohort study conducted in England, insulin therapy, but also therapy with DPP-IV inhibitors have been linked to increased COVID-19 mortality in patients with type 2 diabetes mellitus. The authors of this study discussed that this could be a result of confounding factors, as insulin-treated patients are older, and also DPP-IV inhibitors are more commonly prescribed in older and frail patients [20]. As type 2 diabetic patients with infections such as influenza or pneumonia are at an increased risk of suffering a more severe trajectory of the disease when compared to controls [16,21], bacterial coinfection with SARS-CoV-2 increases the mortality risk dramatically in an intensive care unit setting, especially for patients with diabetes [22]. Hence clarity regarding the role of antidiabetic drugs is of the utmost importance. In our study, the high-risk population for developing pneumonia, namely patients with type 2 diabetes mellitus undergoing insulin therapy, had a lower mean age compared to the control group but had a higher rate of comorbidities when compared to the controls. A possible mechanism behind an increase in pneumonia cases in insulin-treated patients is the immunomodulatory nature of insulin itself. To date, insulin has been shown to exert both pro- and anti-inflammatory effects [23]. On the one hand, insulin promotes T cell proliferation and function by enhancing cellular metabolism [23]—an effect which cannot be seen in T cells in an insulin-resistant state [24]. However, insulin dampens the proinflammatory response due to the presence of lipopolysaccharide (LPS) by preventing LPS-induced production of reactive oxygen species (ROS) [25] and cytokines such as IL-6 or TNF-α, probably by activating the phosphatidylinositol 3-kinase/protein kinase B pathway [26]. Of note, the immunomodulatory effects of insulin seem to be mediated through glucose metabolism [27]. This complex interplay has already been investigated in a laboratory setting. For example, when septic diabetic rats were treated with insulin, an increase in prostaglandin E2 and transforming growth factor β levels could be measured in the rats’ lungs [28]. In our study, we built up different patient groups according to a stable antidiabetic drug therapy of 4 years and we found an increased risk of pneumonia in type 2 diabetic patients treated with insulin. In detail, insulin as monotherapy (OR: 2.07), but also in combination with statins (OR: 2.24) or with statins and metformin (OR: 2.27) was related to an increased risk of pneumonia. We also found a higher risk of being diagnosed with pneumonia in patients treated with a combination of insulin, statins and DPP-IV inhibitors and undergoing combination therapy with sulfonylurea and metformin, demonstrating that insulin and insulin-secreting oral antidiabetic drugs seem to compound the development of pneumonia. Contrary to the association of insulin and pneumonia risk, Yang et al. demonstrated in a retrospective cohort study of 22,638 patients with type 2 diabetes mellitus that patients who had at least 90 days’ stable metformin therapy have a reduced risk of pneumonia [29]. However, the control group in the study by Yang et al. included patients with no further specified antidiabetic drugs except for sulfonylurea monotherapy and the authors did not report their results in comparison to patients without drug therapy. Additionally, the stable antidiabetic therapy was relatively low compared to our study. Beneficial effects on inflammatory markers in animal models [30], and the neutrophil/lymphocyte ratio [31] have been reported for metformin therapy. In the present study, we could not find a reduced risk of pneumonia for different forms of antidiabetic drugs (including metformin monotherapy) in patients with type 2 diabetes mellitus.

We have to report limitations in the present study: First, we had no information as to when antidiabetic treatment was started in the investigated populations. However, a strength of the present study is that all patients investigated had received stable antidiabetic drug therapy over the baseline period of 4 years. In this longitudinal study we cannot report causalities, only relationships. In the present study, we do not have longitudinal data for the new antidiabetic drugs such as sodium-glucose cotransporter-2 inhibitors (SGLT-2 inhibitors) or glucagon-like peptide 1 (GLP-1) agonists, and, therefore, we did not report results about these drugs. Additionally, we had no data on laboratory parameters such as HbA1c and therefore we cannot evaluate the exact status of diabetes control. Thus, our results are limited to ATC- and ICD-codes and hence we cannot characterize the patients according to their laboratory parameters. We cannot discount the possibility that a small number of the investigated patients were diagnosed with type 1 diabetes mellitus, as we do not have data on diabetes antibodies. Another limitation is that we have excluded deceased patients, as we did not know the exact cause of death. Nor can we rule out the possibility that the present study also analyzed patients with secondary diabetes mellitus due to long-term corticosteroid intake, a form of diabetes that is related to a higher rate of infectious diseases. Additionally, a possible higher rate of vaccination coverage has to be considered in insulin treated patients.

In conclusion, we could show that in patients with type 2 diabetes mellitus, insulin therapy is related to an increased risk of being diagnosed with pneumonia and that there is no antidiabetic drug therapy in the present study which is related to a lower risk of being diagnosed with pneumonia. Larger prospective RCTs ought to be conducted to deepen the understanding of the relationship between different antidiabetic drug combinations in type 2 diabetic patients with respiratory tract infections.

## 4. Methods

### 4.1. Dataset and Study Population

We used a dataset containing information on all Austrian diabetes patients (type 2 diabetes mellitus) over a study period of 6 years (2012–2017). The dataset was derived from medical claims data and includes information on 904,032 patients meeting at least one of the following criteria: member of a diabetes disease management program; oral anti-diabetic therapy (ATC codes starting with A10B); insulin therapy (ATC codes starting with A10A and with no recorded diagnosis of ICD-code E10 in any year of the study period); or member of a diabetes risk group (≥4 blood glucose measurements per year or ≥1 HbA1c measurements per year). The yearly datasets include all hospital admissions of such patients as well as their diagnoses (according to the International Classification of Diseases, 10th Revision, ICD codes) and additional information on prescribed medications (ATC codes).

In our study, we used patient-specific information (sex, date of birth, date of death) as well as information on prescribed medications (date, amount and type) and diagnoses (date and type) patients received during hospital admissions. We excluded patients with no known date of birth or sex (n = 3) and patients with known date of death (n = 179,081).

### 4.2. Outcome Diagnoses, Definition of Baseline and Observational Period

The outcome was defined via hospital admissions with any primary or secondary diagnosis of pneumonia with ICD-10 codes J12–J18. The outcome was recorded in the observational period from 2016 to 2017 and is either positive or negative for each individual patient. Patients with any hospitalization with ICD-10 codes J12–J18 during the baseline period (2012 to 2015) were excluded. Additional information on patients (such as hospitalizations, medication or diagnoses for correction) was also recorded in the baseline period (2012 to 2015).

We further divided patients into therapy cohorts and the control cohort.

### 4.3. Definition of Therapy Groups and Controls

The therapy cohort included n = 31,397 patients and was further divided into separate therapy groups based on prescribed ATC codes. Therapy groups contained all antidiabetic medications (ATC codes that start with A10A for insulins, A10BA—biguanides, A10BB—sulfonylureas, A10BC—sulfonamides, A10BD—combination therapies, A10BF—alpha-glucosidase inhibitors, A10BG—thiazolidinediones, A10BH—DPP-IV inhibitors, A10BJ—GLP-1 analoga, A10BK—SGLT2 inhibitors, A10BX—other blood-glucose-lowering drugs) and statins (with ATCs C10AA).

Only patients with continuous therapies were considered, meaning a minimum of 4 prescriptions/year per ATC code throughout the entire baseline period. Patients in therapy groups must have had been continuously prescribed only the specified ATC codes and none of the other ATC codes to avoid bias due to the effect of other medication. There were no therapy groups containing ATCs A10BJ and A10BK. Those ATCs were not prescribed in Austria until 2016 and thus not included in the baseline period. Additionally, therapy groups must have consisted of a minimum of 10 patients; otherwise, they were not considered for further analysis.

The control group (n = 6568) consisted of patients without any prescription of the above ATC codes throughout the baseline period.

### 4.4. Statistical Analysis

We used multiple logistic regression for outcome risk assessment for every therapy group separately, comparing drug therapy to controls without antidiabetic medication. Models were performed only where the number of patients with and without outcomes in the therapy group and control group was >5. In all models, we used age, sex and hospitalization days as covariates. The number of hospitalization days was used as an approximation of patients’ general health status. Therapy was coded as a binary variable with a value of 1 for patients in therapy groups and 0 for patients in the control cohort. The status of outcome event in the observational period was denoted by 1 (if outcome occurred) or 0 (if outcome was not observed = censored). The statistical analysis was performed in Python, mainly using the statsmodels package [32] for the regression models and pandas [33] for data handling.

The regression results showed the estimated odds ratios as well as the 95% confidence intervals for the outcome of pneumonia in hospital admissions. We performed two types of models.
The general model contains all patients and adjusts for sex, age and number of hospital days in baseline period (regression formula: outcome ~ sex+age+hospital days+therapy).The sex-specific model adjusts for age and number of hospital days in the baseline period, with separate regression models for female and male patients (regression formula: outcome ~ age+hospital days+therapy).

For the results displayed, all therapy groups containing ATC code A10BD (combination therapies) were excluded, since it is not specified which medications these patients were prescribed.

## Figures and Tables

**Figure 1 jpm-11-00984-f001:**
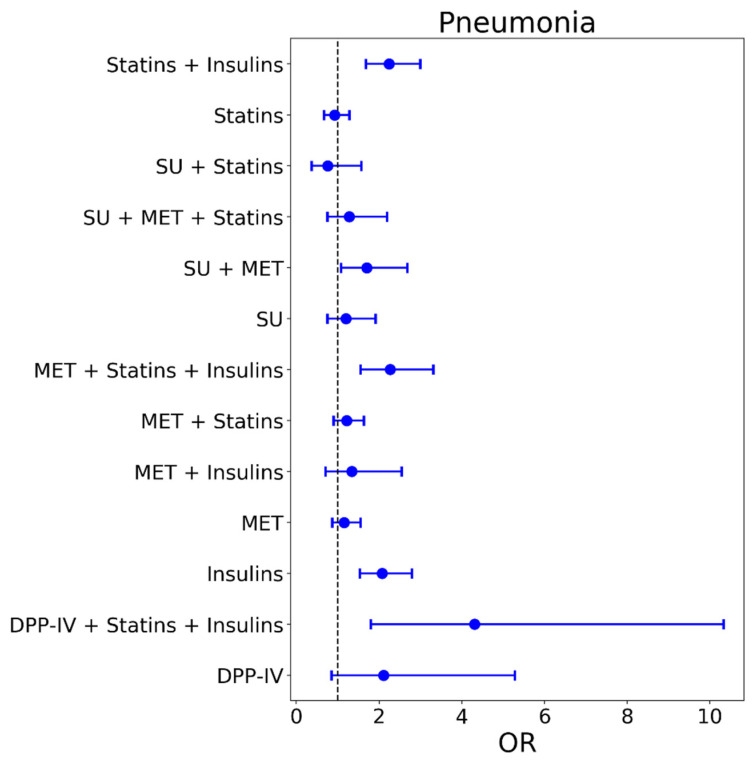
Odds ratios of pneumonia outcome for therapy groups compared to controls.

**Table 1 jpm-11-00984-t001:** Baseline table. All valid therapy groups with number of patients that fulfill the specified criteria and % of patients in therapy groups with observed outcome of pneumonia in 2016–2017. Additional info on mean (±STD) age and sex distribution in therapy groups and controls.

Therapy Group	Number of Patients in Therapy Group	% of Patients with Outcome in Observation Period	Mean Age at End of Baseline Period	Sex Distribution (% Females/% Males)
MET ^1^	6798	1.37	67.83 ± 10.83	49%/51%
MET + Statins	5865	1.52	69.85 ± 9.18	48%/52%
Statins	3872	1.55	73.56 ± 8.55	50%/50%
Insulin	3720	2.45	66.57 ± 9.72	42%/58%
Insulin + Statins	2939	3.27	70.62 ± 8.95	43%/57%
MET + Insulin + Statins	1596	2.69	68.27 ± 8.86	49%/51%
MET + SU ^2^	1122	2.14	70.25 ± 10.36	52%/48%
SU	1091	2.02	74.63 ± 11.19	55%/45%
MET + SU + Statins	882	1.81	71.87 ± 8.62	47%/53%
MET + Insulin	722	1.52	66.50 ± 11.48	56%/44%
SU + Statins	555	1.44	76.38 ± 8.89	54%/46%
MET + TZD ^3^ + Statins	201	0.5	69.06 ± 8.24	49%/51%
DPP-IV ^4^	191	2.62	69.44 ± 10.41	56%/44%
MET + DPP-IV + Statins	189	2.12	69.19 ± 9.10	52%/48%
DPP-IV + Statins	183	1.64	72.47 ± 9.63	53%/47%
MET + TZD	158	1.9	67.51 ± 7.94	49%/51%
DPP-IV + SU	140	2.14	73.63 ± 10.19	49%/51%
DPP-IV + SU + Statins	134	0.75	76.28 ± 8.04	55%/45%
MET + DPP-IV	137	1.46	64.28 ± 9.89	45%/55%
TZD	116	0.86	70.09 ± 9.98	44%/56%
TZD + Statins	105	2.86	70.99 ± 8.96	49%/51%
MET + other-AD ^5^	96	3.12	69.74 ± 9.56	46%/54%
DPP-IV + Insulins + Statins	92	6.52	72.48 ± 7.62	60%/40%
TZD + SU	86	1.16	74.59 ± 9.74	57%/43%
TZD + SU + Statins	78	2.56	75.13 ± 7.01	54%/46%
MET + TZD + SU	70	4.29	70.99 ± 8.71	50%/50%
Other-AD + Statins	62	3.23	73.77 ± 8.28	36%/64%
Alpha-Gluc.-Inh. ^6^	46	2.17	70.35 ± 14.36	65%/35%
Alpha-Gluc.-Inh. + Statins	37	2.7	74.44 ± 10.01	45%/55%
DPP-IV + Insulin	31	3.23	69.78 ± 11.42	72%/28%
Insulin + SU	26	3.85	68.49 ± 11.00	65%/35%
MET + Alpha-Gluc.-Inh.	23	8.7	69.39 ± 13.11	52%/48%
DPP-IV + Other-AD + Statins	13	7.69	67.31 ± 9.99	38%/62%
MET + Alpha-Gluc.-Inh. + Statins	11	9.09	75.18 ± 6.27	73%/27%
DPP-IV + Other-AD	10	10	69.91 ± 11.94	100%/0%
Controls	6568	1.55	72.83 ± 9.96	58%/42%

^1^ Metformin; ^2^ sulfonylureas; ^3^ thiazolidinediones; ^4^ DPP-IV inhibitors; ^5^ other blood-glucose-lowering drugs; ^6^ alpha-glucosidase inhibitors.

**Table 2 jpm-11-00984-t002:** Sex-specific regression results for models with male patients only. MET = metformin, SU = sulfonylureas.

Therapy Group	OR (Confidence Interval)	*p*-Value
MET	1.15 (0.78–1.71)	0.48
MET + Statins	1.31 (0.88–1.94)	0.19
Statins	1.18 (0.79–1.78)	0.42
Insulin	2.02 (1.35–3.03)	<0.001
Statins + Insulin	2.38 (1.61–3.51)	<0.001
MET + Statins + Insulin	2.44 (1.46–4.07)	<0.001
MET + SU	1.53 (0.80–2.91)	0.20
SU	1.61 (0.89–2.91)	0.12
MET + Statins + SU	1.18 (0.57–2.42)	0.66
MET + Insulin	1.28 (0.50–3.33)	0.61
Statins + SU	1.06 (0.45–2.50)	0.90

**Table 3 jpm-11-00984-t003:** Sex-specific regression results for models with female patients only. MET = metformin, SU = sulfonylureas, OR = odds ratio, CI: confidence interval.

Therapy Group	OR (CI)	*p*-Value
MET	1.17 (0.75–1.82)	0.49
MET + Statins	1.09 (0.70–1.71)	0.70
Statins	0.61 (0.35–1.07)	0.09
Insulin	2.17 (1.40–3.37)	<0.001
Statins + Insulin	2.12 (1.37–3.29)	<0.001
MET + Statins + Insulin	2.08 (1.17–3.70)	0.012
MET + SU	1.91 (1.00–3.64)	0.048
SU	0.80 (0.36–1.78)	0.59
MET + Statins + SU	1.45 (0.65–3.23)	0.36
MET + Insulin	1.41 (0.59–3.35)	0.43

## Data Availability

The data was made available for selected research partners under strict data protection regulations. For more information on permission to access the data contact P.K.

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
