# Peer review of "Insulin as Monotherapy and in Combination with Other Glucose-Lowering Drugs Is Related to Increased Risk of Diagnosis of Pneumonia: A Longitudinal Assessment over Two Years"

_jpm, 2021, doi:10.3390/jpm11100984_

Round 1

Reviewer 1 Report

The manuscript is well presented and adds new evidence to the association of insulin with pneumonia. Nevertheless, I have some doubts about some methodological aspects:

  • Why the authors did not adjust in the logistic regression for comorbidities and years of evolution of the diabetes? I think those variables need to be included in the adjustment in order to try to compare similar patients. How can you be sure that patients with higher risk of pneumonia were not "more severe" patients, with more comorbidities and/or a longer-evolution diabetes, with more macrovascular and microvascular complications developed during time?
  • Why are statins included as study drugs if the objective is to assess diabetes treatments? 
  • In Figure 1, the authors report higher risk of pneumonia with some combinations not including insulin, so I do not think that the pneumonia risk can be only attributed to insulin. 

Author Response

We thank the reviewer for his positive comments.

Comment:

Why the authors did not adjust in the logistic regression for comorbidities and years of evolution of the diabetes? I think those variables need to be included in the adjustment in order to try to compare similar patients. How can you be sure that patients with higher risk of pneumonia were not "more severe" patients, with more comorbidities and/or a longer-evolution diabetes, with more macrovascular and microvascular complications developed during time?

Response: Although we tried to adjust for such “more severe” patients by including the number of days spent in hospital care in the logistic regression analyses, it might be useful to test the effect of specific diabetic complications as covariates in the model. Additionally, we have now adjusted for the listed comorbidities (from the supplementary table) and alcohol abuse and nicotine dependence (ICD codes F10 and F17) in the logistic regression. In this additional analysis the results did not change qualitatively. We now state this in the results section as well as the statistical analysis subsection in the methods section.

Unfortunately, we are not able to adjust for the years of evolution of diabetes since it is unknown how long patients were diabetic before the beginning of the data (2012). While this is a valid limitation to our study, we seek to build homogeneous cohorts with a continuous 4-year medication therapy.

 Comment:

Why are statins included as study drugs if the objective is to assess diabetes treatments?

Response: Due to the guidelines of the ESC/EAS 2019 on the treatment of hyperlipidemia, statins are a common therapy in patients with diabetes mellitus type 2 and therefore we included them in the analyses. As a supplement to our presented study, we have run the same analysis without the inclusion of statin medication, without a qualitative change in resulting effects. It still appears that therapy groups that include insulin (Insulins + DPP-IV, Insulins + MET) or insulin monotherapy are at higher outcome risk compared to some other therapy groups. Additionally, the groups of MET and MET + SU + TZD show increased OR.

Comment:

In Figure 1, the authors report higher risk of pneumonia with some combinations not including insulin, so I do not think that the pneumonia risk can be only attributed to insulin.

Response: We are only able to compare therapy groups to a control cohort that is not prescribed with any of the studied medication codes. It is correct that we cannot only attribute the higher risk to patients with insulin prescriptions, although most of the therapy groups containing insulins show significant increased OR while other therapies have lower OR (no effect, close to one).

Reviewer 2 Report

The manuscript from Leutner and colleagues reported an increased risk of pneumonia diagnosis among type 2 diabetes mellitus patients treated by insulin, insulin +statins, insulin+ metformin+statins as well as those treated by DPP4 inhibitors+ insulin+ statins.

Several minor remarks can be made:

1/ scope of the journal: This epidemiological manuscript did not fit to primary topics of interest of the journal:

Omics-based studies of disease risk, disease prognosis, and response to therapy

Biomarker identification and application

Drug target discovery and integration with individualized therapy

Pharmacogenomics – genetics and biochemistry of drug uptake, action, and metabolism

Prediction-based drug safety

Novel therapeutics: genetic-based, nanotechnology, targeting strategies

Advancements in Technologies and Resources

Genetic testing, sequencing technologies, and molecular diagnostics

Expression analysis, metabolomics, proteomics

Microbiomes

Genome-wide association studies

Bioinformatics and health informatics

Biostatistics

Personalization of omics-based non-drug related health interventions

Regulation and bioethics in personalized medicine

2/ title : “increased risk of diagnosis of pneumonia” should be preferred to “increased risk of pneumonia”  and “over two years” should be preferred to “over six years”(see below)

3/ introduction: the introduction focused on the potential detrimental role of antidiabetic drug in the occurrence of pneumonia. In this context, we do not understand why the authors added statins (which is not primarily an antidiabetic drug) among the “therapy groups”. I would suggest the authors to speak about the potentially beneficial role of statins onto the pneumonia outcome (Mortensen et al Clin infect Dis 2012) in their introduction if they want to justify “statins” among the “therapy groups”.

4/methods: several bias here should be mentioned in the “limits “part of the discussion section:

  • The possibility that type I diabetes mellitus could not have been excluded when E10 ICD code had been recorded before the study period. That is all the more important because insulin is an early treatment during type I diabetes mellitus, without the same prognostic meaning of a long lasting disease in an older patient.
  • A bias due to the exclusion of deceased patients, which could have died from severe pneumonia
  • A selection bias linked to the exhaustivity of ICD or ATC codage. Moreover, no clinical or biological data allowed confirming the reality of pneumonia diagnosis. That is the reason why the title should be changed (see above)
  • The outcome pneumonia diagnosis yes/no has only been recorded between 2016 and 2017 and not during the whole 6 years (see above the title section)
  • Secondary diabetes mellitus due to long term corticosteroids treatment has not been considered, whereas this kind of diabetes mellitus has probably higher trends to bacterial infections because of the immunosuppressive role of corticosteroids.
  • Alcohol consumption and tobacco smoking which have been associated with the occurrence of pneumococcal infections (the most frequent bacterial cause of pneumonia) have not been considered (Kostas et al, Open forum infectious diseases 2019)

5/ results:

  • Why “DPP4 inhibitors + statins + insulin” therapy group is not found among tables 2 and 3?
  • Why the authors did not integrate the numbers of comorbidities shown in supplementary table 1 in their multivariate model?

6/discussion:

-page 7 : line 1 and 2. The absence of laboratory data such as HbA1c should be discussed in line with reference 8.

I would add a major remark concerning results and discussion.

Despite a thorough epidemiological work, I do not feel that we could incriminate antidiabetic treatment in the occurrence of pneumonia, when reading these results.

First, if statins are likely to be associated with a better pneumonia outcome (Mortensen et al Clin infect Dis 2012), it would seem unlikely that statins are associated with pneumonia. To support this consideration, the percentage of patients with outcome in observation period is the same in statins and control groups (table 1). Therefore, what is the interest of separating groups according to statins? If the authors want to keep statins, these results should be shown first as non-significant, before rebuilding therapy groups (Met and Met + statins should become a new MET group, Met + insulin and Met + insulin + statins should become a new Met + insulin group). Would the results be the same when groups are gathered like this without the influence of statins?

Secondly, as stated above, the number of comorbidities should be added in the multivariate model because the numbers of comorbidities is associated with a worse prognosis during non-meningitis pneumococcal infections (Kostas et al, Open forum infectious diseases 2019). This is all the more important because insulin could be associated with a longer evolution of diabetes mellitus among type 2 diabetes mellitus patients and possibly with a higher number of comorbidities. Moreover, the interest of removing statins from the primary results would be that we did not know how to interpret statins in this population. Are statins associated with stroke or ischemic heart disease? Are statins associated with silent hypercholesterolemia or with the presence of a considerate general practitioner? As the authors did for DPP4 inhibitors which is considered as a marker of frailty (line 129 p6), the authors should interpret what factor could be collinearly associated with insulin before suggesting a potential role of insulin itself on the occurrence of pneumonia.

Author Response

We thank the reviewer for his positive comments.

Reviewer’s comment:

1/ scope of the journal: This epidemiological manuscript did not fit to primary topics of interest of the journal

Response: The aim of the present study was to investigate the relationship between specific antidiabetic drug combinations and the diagnosis of pneumonia in order to increase the understanding of personalized medicine in the high-risk population of type 2 diabetic patients.

Reviewer’s comment:

2/ title: “increased risk of diagnosis of pneumonia” should be preferred to “increased risk of pneumonia” and “over two years” should be preferred to “over six years”(see below)

Response: We have changed the wording in accordance with the reviewers’ comment.

Reviewer’s comment:

3/ introduction: the introduction focused on the potential detrimental role of antidiabetic drug in the occurrence of pneumonia. In this context, we do not understand why the authors added statins (which is not primarily an antidiabetic drug) among the “therapy groups”. I would suggest the authors to speak about the potentially beneficial role of statins onto the pneumonia outcome (Mortensen et al Clin infect Dis 2012) in their introduction if they want to justify “statins” among the “therapy groups”.

Response: We thank the reviewer for this important comment. We have now added a paragraph to statins and pneumonia in the introduction section. Additionally, we also analysed the results without statins and couldn’t find a qualitative change.

Reviewer’s comment:

4/methods: several bias here should be mentioned in the “limits “part of the discussion section:

  • Comment: The possibility that type I diabetes mellitus could not have been excluded when E10 ICD code had been recorded before the study period. That is all the more important because insulin is an early treatment during type I diabetes mellitus, without the same prognostic meaning of a long lasting disease in an older patient.
  • Response: In line with the reviewers’ comment, we have stated in the limitations section that we have no data on diabetes antibodies and cannot rule out the possibility  that patients with diabetes mellitus type 1 are also included in the analysis.
  • Comment: A bias due to the exclusion of deceased patients, which could have died from severe pneumonia
  • Response: We have added to the limitations section that we have excluded deceased patients, as we did not know the exact cause of death.
  • Comment: A selection bias linked to the exhaustivity of ICD or ATC codage. Moreover, no clinical or biological data allowed confirming the reality of pneumonia diagnosis. That is the reason why the title should be changed (see above)
  • Response: In line with the reviewers’ comment, we have changed the title. Additionally we state that we had no data on laboratory parameters such as HbA1c and therefore we cannot evaluate the exact status of diabetes control
  • Comment: The outcome pneumonia diagnosis yes/no has only been recorded between 2016 and 2017 and not during the whole 6 years (see above the title section)
  • Response: We have revised the title in accordance with the reviewers’ comment.
  • Comment: Secondary diabetes mellitus due to long-term corticosteroids treatment has not been considered, whereas this kind of diabetes mellitus probably has higher trends for bacterial infections because of the immunosuppressive role of corticosteroids.
  • Response: We have added a specific paragraph to the limitations section.
  • Comment: Alcohol consumption and tobacco smoking, which have been associated with the occurrence of pneumococcal infections (the most frequent bacterial cause of pneumonia), have not been considered (Kostas et al, Open forum infectious diseases 2019)
  • Response: In line with the reviewers’ comment, we have controlled our results on alcohol abuse (ICD F10), nicotine dependence (ICD F17) and other comorbidities from the supplementary table. Hence the results did not change qualitatively.

Reviewer’s comment:

5/ results: Why “DPP4 inhibitors + statins + insulin” therapy group is not found among tables 2 and 3?

Response: While Figure 1 of the paper presents the OR of the non-sex-specific study and includes the therapy group “DPP-IV inhibitors + statins + insulin”, Table 2 and Table 3 show the results of the sex-specific analysis. In the sex-specific analysis, case numbers for specific therapy groups (like those mentioned above) were too low and were therefore not suitable for regression analysis.

Reviewer’s comment:

Why the authors did not integrate the numbers of comorbidities shown in supplementary table 1 in their multivariate model?

Response: In response to the reviewer’s comment, we performed an additional test including several diabetic complications in the regression model (specified as diagnosed ICD codes) from supplementary table 1 as well as alcohol abuse (ICD code F10) and nicotine dependence (ICD code F17) to adjust for the effects of multimorbid conditions. We did not observe any qualitative changes in results. In our study, we use the number of hospital days as a proxy for the health status of patients instead of the number of comorbidities.

Reviewer’s comment:

6/discussion:

-page 7 : line 1 and 2. The absence of laboratory data such as HbA1c should be discussed in line with reference 8.

I would add a major remark concerning results and discussion.

Despite a thorough epidemiological work, I do not feel that we could incriminate antidiabetic treatment in the occurrence of pneumonia, when reading these results.

Response: In line with the reviewer’s comment, we have now discussed the absence of laboratory data in more detail.

Reviewer’s comment:

First, if statins are likely to be associated with a better pneumonia outcome (Mortensen et al Clin infect Dis 2012), it would seem unlikely that statins are associated with pneumonia. To support this consideration, the percentage of patients with outcome in observation period is the same in statins and control groups (table 1). Therefore, what is the interest of separating groups according to statins? If the authors want to keep statins, these results should be shown first as non-significant, before rebuilding therapy groups (Met and Met + statins should become a new MET group, Met + insulin and Met + insulin + statins should become a new Met + insulin group). Would the results be the same when groups are gathered like this without the influence of statins?

Response: In accordance with the reviewer’s comment, we have run the same analysis without the inclusion of statins so that there was no explicit dependence of the outcome on statin prescriptions. Qualitatively, we still observe similar effects – therapy groups containing insulins or insulin monotherapy show increased outcome OR. We have added a statement about this robustness test in the results section.

Reviewer’s comment:

Secondly, as stated above, the number of comorbidities should be added in the multivariate model because the numbers of comorbidities is associated with a worse prognosis during non-meningitis pneumococcal infections (Kostas et al, Open forum infectious diseases 2019). This is all the more important because insulin could be associated with a longer evolution of diabetes mellitus among type 2 diabetes mellitus patients and possibly with a higher number of comorbidities. Moreover, the interest of removing statins from the primary results would be that we did not know how to interpret statins in this population. Are statins associated with stroke or ischemic heart disease? Are statins associated with silent hypercholesterolemia or with the presence of a considerate general practitioner? As the authors did for DPP4 inhibitors which is considered as a marker of frailty (line 129 p6), the authors should interpret what factor could be collinearly associated with insulin before suggesting a potential role of insulin itself on the occurrence of pneumonia.

Response: In accordance with the reviewer’s comment, we have tested the same analysis without the inclusion of statin medication without qualitative change to the results. Therapy groups that include insulin (Insulins + DPP-IV, Insulins + MET) or insulin monotherapy are still at higher risk compared to other therapy groups. Additionally, the groups of MET and MET + SU + TZD show increased OR as well. We added this to the results section in the subsection for baseline characteristics.

Round 2

Reviewer 2 Report

The authors have answered to all my comments.

If authors could, I would suggest the authors to detail in the abstract that pneumonia occurence have been assessed during 2 years in patients previously screened for antidiabetic treatment during the 4 previous years, in order to avoid discrepancies between title and abstract.

My last remark would be to speak in the discussion about the further importance of high vaccination coverage (especially antipneumococcal) among the patients treated by insulin, taken into account their own results.

Author Response

We thank the reviewer for his positive comments. In accordance with the reviewer's comment we have now revised the abstract and added a specific paragraph to the discussion section. 
